XYA-2: a marine-derived compound targeting apoptosis and multiple signaling pathways in pancreatic cancer

Guan Xiaoqing 1 2 guanxq1991@outlook.com
Li Yun 1
Guan Xiaodan 1
Fan Linfei 1
Ying Jieer 1 2 jieerying@aliyun.com
1 Zhejiang Cancer Hospital, Hangzhou Institute of Medicine (HIM), Chinese Academy of Sciences , Zhejiang , China
2 Key Laboratory of Prevention, Diagnosis and Therapy of Upper Gastrointestinal Cancer of Zhejiang Province , Zhejiang , China
Yan Yuanliang
Electronic publication date: 2024 Jan 16
Publication date: 2024
Volume: 12
Electronic Location ID: e16805
Received 2023 Oct 20; Accepted 2023 Dec 28
Copyright: © 2024 Guan et al.
Copyright year: 2024
Copyright holder: Guan et al.
License: This is an open access article distributed under the terms of the Creative Commons Attribution License, which permits unrestricted use, distribution, reproduction and adaptation in any medium and for any purpose provided that it is properly attributed. For attribution, the original author(s), title, publication source (PeerJ) and either DOI or URL of the article must be cited.
License URL: https://creativecommons.org/licenses/by/4.0/

Keywords: Pancreatic cancer, XYA-2, Apoptosis, Cell proliferation, Pancreatic cancer treatment

Funding: Traditional Chinese Medicine Science and Technology Program of Zhejiang Province 2023ZR003 Prevention, Diagnosis, and Therapy of Upper Gastrointestinal Cancer of Zhejiang Province 2022E10021 This research was supported by grants from the Traditional Chinese Medicine Science and Technology Program of Zhejiang Province (2023ZR003), and the Key Laboratory of Prevention, Diagnosis, and Therapy of Upper Gastrointestinal Cancer of Zhejiang Province (2022E10021). The funders had no role in study design, data collection and analysis, decision to publish, or preparation of the manuscript.

==============================
Background

Pancreatic cancer is a highly aggressive and fatal disease with limited treatment options and poor prognosis for patients. This study aimed to investigate the impact of XYA-2 {N-(3,7-dimethyl-2,6-octadienyl)-2-aza-2-deoxychaetoviridin A}, a nitrogenated azaphilon previously reported from a deep-sea-derived fungus on the progression of pancreatic cancer cells.

Methods

The inhibitory effects of XYA-2 on cell proliferation, clonogenic potential, cell cycle progression, apoptosis, migration, and invasion were assessed using various assays. The CCK-8 assay, clone formation assay, flow cytometry assay, wound healing assay, and transwell assay were employed to evaluate cell proliferation, clonogenic potential, cell cycle progression, apoptosis, migration, and invasion, respectively. Moreover, we employed RNA-seq and bioinformatics analyses to uncover the underlying mechanism by which XYA-2 influences pancreatic cancer cells. The revealed mechanism was subsequently validated through qRT-PCR.

Results

Our results demonstrated that XYA-2 dose-dependently inhibited the proliferation of pancreatic cancer cells and induced cell cycle arrest and apoptosis. Additionally, XYA-2 exerted a significant inhibitory effect on the invasion and migration of cancer cells. Moreover, XYA-2 was found to regulate the expression of genes involved in multiple cancer-related pathways based on our RNA-seq and bioinformatics analysis.

Conclusion

These findings highlight the potential of XYA-2 as a promising therapeutic option for the treatment of pancreatic cancer.

Introduction

Pancreatic cancer stands out as a pervasive and exceptionally deadly malignancy among humans. In 2020 alone, it accounted for a staggering 466 thousand new fatalities, solidifying its position as the seventh leading cause of cancer-related deaths across 185 countries (Sung et al., 2021). According to the Global Cancer Observatory (GLOBOCAN), the 5-year overall survival rate for pancreatic cancer, based on diagnoses made between 2012 and 2018, is dishearteningly low at just 12% (Siegel et al., 2023). This stark reality is largely attributed to the inherent challenges associated with early-stage detection of pancreatic cancer, resulting in the majority of patients receiving diagnoses at advanced stages. Consequently, this delayed diagnosis contributes significantly to the overall poor prognosis associated with this formidable disease.

The primary treatment approach for pancreatic cancer involves surgical resection combined with postoperative chemotherapy and radiotherapy. Chemotherapy serves as the standard adjuvant treatment after surgery, with different chemotherapeutic agents utilized depending on the specific gene mutations present in pancreatic cancer. Tarceva (Erlotinib) and Lynparza (Olaparib) have both been approved by the FDA as targeted therapies for pancreatic cancer. Erlotinib is an inhibitor of the epidermal growth factor receptor (EGFR) tyrosine kinase. It is recommended for use in combination with first-line treatment for patients diagnosed with locally advanced, unresectable, or metastatic pancreatic cancer (Sohal et al., 2016). Olaparib is a poly (ADP-ribose) polymerase (PARP) inhibitor. Olaparib is indicated for the maintenance treatment of adult patients with deleterious or suspected deleterious germline BRCA mutations (gBRCAm) metastatic pancreatic adenocarcinoma whose disease has not progressed after at least 16 weeks of a first-line platinum-based chemotherapy regimen (Kindler et al., 2022). Frequent occurrences of KRAS mutations characterize the pancreatic cancer (Zhu et al., 2022), with G12V and G12D mutations being the most prevalent, collectively representing approximately 90% of cases. A noteworthy advancement in targeted therapy is MRTX1133 (Wang et al., 2022), demonstrating potential in inhibiting KRAS G12D and introducing novel avenues for pancreatic cancer treatment. In contrast, KRAS G12C mutations are less common, manifesting in only 1% to 2% of pancreatic cancer cases (Luo, 2021). Promising developments include Sotorasib (AMG510) (Strickler et al., 2023), and Adagrasib (MRTX849) (Fell et al., 2020), both emerging as effective KRAS G12C inhibitors. Currently, the only immunotherapy drug approved for advanced pancreatic cancer is PD-1 monoclonal antibody. It is endorsed for the treatment of patients exhibiting high microsatellite instability (MSI-H) or mismatch repair deficiency (dMMR) biomarkers, as well as a high tumor mutation burden (Maio et al., 2022). However, despite these advancements, the prognosis remains unsatisfactory, underscoring the pressing need for novel drug development to alleviate the burden imposed by pancreatic cancer.

In a previous study, we identified XYA-2 (Fig. 1A) as a novel inhibitor of STAT3 (signal transducer and activator of transcription 3) (Guan et al., 2023). In vitro assays elucidated XYA-2’s capacity to impede STAT3 phosphorylation in gastric cancer cells, resulting in diminished cellular activity and suppressed cell proliferation. In vivo experiments showcased XYA-2’s ability to inhibit tumor growth and metastasis in gastric cancer mouse models, exhibiting minimal toxicity. Beyond its efficacy in gastric cancer, STAT3 has emerged as a promising target for pancreatic cancer treatment (Chen et al., 2023; He et al., 2022). The activation of STAT3 signaling in pancreatic cancer is implicated in cell survival promotion, apoptosis inhibition (Glienke, Hausmann & Bergmann, 2011), and the upregulation of MMP7 expression, fostering cancer growth and metastasis (Fukuda et al., 2011). STAT3 also plays a role in shaping the pancreatic cancer microenvironment, contributing to tumor progression (Yu, Pardoll & Jove, 2009). Genetic deletion of STAT3 in pancreatic epithelial cells has demonstrated a reduction in tumorigenesis in KRAS G12D mouse models (Lesina et al., 2011). Notably, STAT3 inactivation characterizes the normal pancreas (Lee & Hennighausen, 2005), highlighting its dispensability for pancreatic development and hemostasis. Motivated by the pivotal role of the STAT3 pathway in pancreatic cancer, we conducted this study to investigate XYA-2’s impact on pancreatic cancer cells and unravel the associated mechanisms.

Figure 1 XYA-2 inhibits the proliferation of pancreatic cancer cells.

(A) Chemical structure of XYA-2. (B and C) Viability of MIA-PaCa2, PANC-1 and PANC0203 cells treated with XYA-2 evaluated by 24-h, 48-h, and 72-h CCK8 assays. (D) Clonogenicity of MIA-PaCa2 and PANC-1 cells treated with DMSO or XYA-2.

Our findings herein demonstrated that XYA-2 effectively inhibited the activity of pancreatic cancer cells in a concentration-dependent manner. Furthermore, this study suggests that the treatment of pancreatic cancer with XYA-2 may involve the modulation of multiple proteins and signaling pathways. Targeting these specific proteins and signaling pathways may provide novel insights and potential therapeutic avenues for the clinical management of pancreatic cancer.

Materials and Methods

Cell culture

PANC-1 and MIA-PaCa2 cell lines were obtained from the American Type Culture Collection. The cells were cultured in DMEM complete medium supplemented with 10% fetal bovine serum (FBS; GIBCO, Grand Island, NY, USA) and 100 U/mL penicillin-streptomycin solution (Biosharp, Hefei, China). All cell cultures were maintained at a temperature of 37 °C in a 5% CO2 incubator.

Acquisition of XYA-2

XYA-2 (N-(3,7-Dimethyl-2,6-octadienyl)-2-aza-2-deoxychaetoviridin A) was prepared by the same method as previously reported (Guan et al., 2023; Wang et al., 2020) and its purity was higher than 98%. Briefly, XYA-2 was extracted by extracting the fermentation broth, followed by purification by chromatography and preparative HPLC.

Cell viability assay

Cell viability was assessed using the Cell Counting Kit-8 (CCK-8) assay. PANC-1 and MIA-PaCa2 cells were seeded at a density of 5 × 103 cells per well in 96-well plates overnight. Different concentrations of XYA-2 included 1.56, 3.25, 6.25, 12.5, 25, 50, or 100 μM, were added for 24- and 72-h treatment periods. Following the treatment, 10 μL of CCK-8 reagent (Biosharp, Hefei, China) was added to each well, and the absorbance was measured at 450 nm after a 2-h incubation period with the Spark multimode microplate reader (TECAN, Grödig, Austria).

The principles guiding the selection of concentrations for the following in vitro experiments assessing the anti-tumor activity of XYA-2 are as follows: Integrating the goals of various experiments with considerations for cell density, drug exposure duration, and established IC50 values, we opted to assess a range of concentrations, including low, medium, and high. This approach aims to establish a dose-response relationship and comprehend the XYA-2’s impact on tumor cells across a spectrum of concentrations. If high concentrations result in extensive cell death, a reduction in concentration is considered to facilitate the evaluation of experimental parameters.

Colony formation assay

Data were collected following established procedures (Yu, 2022 #43). Briefly, PANC-1 and MIA-PaCa2 cells were seeded in six-well plates at a density of 2,000 cells per well. Following adherence, cells were treated with either 0 or 2 µM XYA-2 for 24 h. Subsequently, the medium was replaced, and cells were cultured at 37 °C for 10–14 days. Colonies were fixed with 4% paraformaldehyde and stained with 0.2% crystal violet. Quantification of visible colonies was conducted using ImageJ software.

Cell cycle arrest assay

PANC-1 and MIA-PaCa2 cells were seeded at 2 × 106 cells per dish. per 10-cm dish overnight and treated with either DMSO (control) or XYA-2 at 2 or 5 μM for 24 h. Then the cells were trypsinized, centrifuged at 1,000 rpm for 5 min at room temperature, and washed three times with 1 mL pre-chilled 75% ethanol. Subsequently, the cells were fixed overnight at −20 °C. Following fixation, the cells were re-suspended and stained with 500 µL PI/RNase staining buffer at room temperature for 30 min. The Attune NxT Flow Cytometer (Thermo Fisher Scientific, Waltham, MA, USA) was employed to detect the stained cells, and the results were analyzed using FlowJo software.

Apoptosis assay

PANC-1 and MIA-PaCa2 cells were seeded at 1 × 106 cells per 6-cm dish overnight and treated with either DMSO (control) or XYA-2 at 5 or 10 μM for 24 h. Then the cells were re-suspended in 500 μL of 1X binding buffer. Subsequently, 5 μL of fluorescein isothiocyanate-coupled annexin and 5 μL of propidium iodide were sequentially added to the cell suspension and incubated for 15 min in the dark at room temperature. The Attune NxT Flow Cytometer (Thermo Fisher Scientific, Waltham, MA, USA) was used to analyze the fraction of apoptotic cells. The data analysis categorized the results as follows: Q1 (top left): necrotic cells, Q2 (upper right): late apoptotic cells, Q3 (bottom right): early apoptosis, and Q4 (lower left): cells that did not undergo apoptosis. The apoptosis rate was calculated as the sum of the percentages from Q2 and Q3 quadrants.

Western blot assay

Human pancreatic cancer cell lines PANC-1 and MIA-PaCa2 were cultured to 40–50% confluency. Treatment with the compound XYA-2 was performed at concentrations of 0, 5, 10, or 20 μM for 24 h. Following treatment, cells were washed with phosphate-buffered saline and lysed using RIPA buffer containing protease inhibitors. The resulting cell lysates were centrifuged, and the supernatants were collected to obtain total protein. Western blot analysis was carried out by separating equal amounts of protein using SDS-PAGE, followed by transfer to a polyvinylidene fluoride membrane. The membrane was then blocked and incubated with specific antibodies targeting Caspase 3, Cleaved Caspase 3, PARP, Cleaved PARP, Stat3, and Phospho-Stat3 (Tyr705). Immunoreactivity was detected using chemiluminescence, and Western blot results were quantified using an image analysis system. The antibodies used in this study were as follows: Caspase 3: Cell Signaling Technology (CST), #9662f; Cleaved Caspase 3: Proteintech, #19677-1-AP; PARP: CST, #9532; Cleaved-PARP: CST, #5625; Stat3: CST, #12640; Phospho-Stat3 (Tyr705): CST, #9145.

Wound healing assay

Upon reaching 80–90% confluency, PANC-1 and MIA-PaCa2 cells were subjected to scratch wound assays. Wounds were generated by scraping the cell monolayers, followed by treatment with various concentrations of XYA-2 compound (0, 2, 5, or 10 μM). The cells were observed using the CKX53 Inverted Microscope (Olympus, Tokyo, Japan) and imaged at 0, 24, and 36 h after scratching to evaluate cell migration at the edges of the scratch area.

Transwell migration assay

A total of 2 × 105 cells in 200 μL of serum-free media, along with 0, 2, or 5 μM XYA-2 were seeded into the upper chamber. The lower chamber was filled with 600 μL of DMEM containing 10% FBS. After incubating at 37°C for 48 h, the cells on the upper membrane were removed using cotton swabs. The cells that had migrated to the lower membrane were fixed with 4% formaldehyde and stained with 2.5% crystal purple. Subsequently, the invaded cells were counted in five different fields under the CKX53 Inverted Microscope (Olympus, Grödig, Japan). The stained cells were observed and photographed using an inverted microscope. Cells that had migrated to the underside of the gel layer appeared as stained spots at the bottom of the chamber. To quantify the extent of migration, the stained spots could be counted using ImageJ software.

RNA-sequencing and bioinformatics analysis

The RNA-seq samples was prepared as previously described (Guan, 2023 #42). Specifically, PANC-1 and MIA-PaCa2 cells were seeded in 10 cm culture dishes at a density of 2 × 106 cells per dish. Experimental groups, comprising a control treated with DMSO and another treated with 10 µM XYA-2 for 24 h, each included three biological replicates. Post-incubation, RNA extraction was performed using Trizol, and RNA-seq data generation was conducted by LC-Bio Technology Co., Ltd. (Hangzhou, China). For the construction of the RNA-seq transcriptome library, 1 µg of total RNA from each of the three biological replicates was utilized, employing the TruSeq TM RNA sample preparation Kit from Illumina (San Diego, CA, USA). The acquired data were subsequently meticulously analyzed using the free online platform provided by Lianchuan Cloud Platform. Differential expression analysis was performed using the DESeq package in the R platform, utilizing the default configuration. Furthermore, gene set enrichment analysis (GSEA) was carried out using hallmark gene sets and Kyoto Encyclopedia of Genes and Genomes (KEGG) gene sets from MSigDB.

RNA extraction, cDNA synthesis, and quantitative real-time PCR

Pancreatic cancer cells were treated with different concentrations of XYA-2 for 24 h, with DMSO treatment used as a control. Each group consisted of three biological replicates. Total RNA was extracted using the protocol provided by the manufacturer of a commercially available RNA extraction kit (YiShan Biotechnology Co. LTD, Shanghai, China). DNase I treatment (RNeasy MinElute Cleanup Kit; QIAGEN, Hilden, Germany) was used to remove and assess the DNA contamination from RNA samples. The quality control of RNA was conducted by a NanoDrop spectrophotometer (ND-2000; Thermo Fisher Scientific, Waltham, MA, USA). The values of A260/A280 of all RNA samples were between 1.8 and 2.0. The Agilent 2100 Bioanalyzer instrument was used to calculate the RNA integrity number (RIN). Inhibition of reverse-transcription activity was checked by dilution of the sample.

For cDNA synthesis, 2 μg of total RNA was reverse-transcribed into complementary DNA (cDNA) using a reverse transcription kit (YiShan Biotechnology Co. LTD, Shanghai, China). The cDNA synthesis reaction was carried out at 37 °C for 60 min, followed by thermal inactivation at 95 °C for 5 min. The synthesized cDNA samples were then subjected to real-time quantitative polymerase chain reaction (qRT-PCR) using specific primers designed for the target genes. The primers used in this study are listed in Table 1. The total reaction volume was 20 μL, consisting of 10 μL SYBR Green Master Mix (QP002; ESscience Biotech Co. LTD, Shanghai, China), 0.4 μL forward primer, 0.4 μL reverse primer, 2 μL cDNA template, and ddH2O added to reach a final volume of 20 μL. The final primer concentration was 200 nM.

Table 1 Primers used for qRT-PCR.

Gene	Primer direction	Primer sequence (5′–3′)	
PLK1	Forward	AAAGAGATCCCGGAGGTCCTA	
PLK1	Reverse	GGCTGCGGTGAATGGATATTTC	
SPC24	Forward	GCCTTCCGCGACATAGAGG	
SPC24	Reverse	CCTGCTCCTTCGCATTGAGA	
TOP2A	Forward	TGGCTGTGGTATTGTAGAAAGC	
TOP2A	Reverse	TTGGCATCATCGAGTTTGGGA	
KIF2C	Forward	CTGTTTCCCGGTCTCGCTATC	
KIF2C	Reverse	AGAAGCTGTAAGAGTTCTGGGT	
KIF4A	Forward	AGCTTCTTTAATCCCGTCTGTG	
KIF4A	Reverse	GGCCAGAGCCCGTTTCTTT	
MKI67	Forward	ACGCCTGGTTACTATCAAAAGG	
MKI67	Reverse	CAGACCCATTTACTTGTGTTGGA	
GAPDH	Forward	ACAACTTTGGTATCGTGGAAGG	
GAPDH	Reverse	GCCATCACGCCACAGTTTC	

qPCR was conducted as previously described (Qi, 2022 #44) utilizing the CFX96 Touch Real-Time PCR Detection System (Bio-Rad, Hercules, CA, USA). In brief, the reaction conditions comprised an initial pre-denaturation at 95 °C for 5 min, succeeded by 40 cycles of denaturation at 95 °C for 10 s and annealing/extension at 60 °C for 30 s. Melt curve analysis was carried out to confirm amplicon specificity. The amplification products had lengths ranging from 110 to 300 bp. To ensure accurate normalization, GAPDH was used as the housekeeping gene and normalizer. All experiments were performed with three technical replicates. The relative gene expression levels were determined using the 2−ΔΔCt method, with PCR efficiencies > 95% and the R2 was > 0.990. The linear dynamic range, Cq variation at the lower limit, and confidence intervals throughout the range were assessed. Limit of detection for each assay were evaluated. Additionally, the qPCR analysis program used, including its source and version, was specified. The Cq method was employed for determination, and outlier identification and disposition methods were described. Results of negative template controls (NTCs) were reported. Repeatability (intra-assay variation) and reproducibility (inter-assay variation measured as %CV) were analyzed. Power analysis was conducted to determine the sample size.

Statistical analysis

The statistical analysis in this study is performed using GraphPad Prism 9 for macOS (Version 9.4.1; GraphPad Software, La Jolla, CA, USA). The data analysis was expressed as mean ± standard error of the mean (SEM). Student’s t-test was used to determine the significant differences between experimental groups and control groups. Statistical significance was considered if P < 0.05.

Results

XYA-2 inhibits the proliferation of pancreatic cancer cells

To investigate the impact of XYA-2 on pancreatic cancer cells, a CCK-8 assay was initially conducted. The results from the CCK-8 assay revealed that XYA-2 effectively suppressed the activity of pancreatic cancer cells after 24 and 72 h of treatment (Figs. 1B and 1C). Moreover, the IC50 values at 24 h were determined to be 13.62, 21.84, and 16.02 μM for MIA-PACA2, PANC-1, and PANC0203, respectively. At 48 h, these values were 3.148, 8.61, and 8.174 μM, and at 72 h, the IC50 values were found to be 1.141, 1.84, and 3.802 μM for MIA-PaCa2, PANC-1, and PANC0203, respectively. These findings suggest that XYA-2 exhibits concentration-dependent inhibition of pancreatic cancer cell activity. In light of these observed results, we have made the strategic decision to prioritize MIA-PaCa2 and PANC-1 for subsequent downstream experiments.

Furthermore, a colony formation assay was performed to assess the effect of XYA-2. We chose a concentration of 2 μM for the clonogenic formation experiment, which was slightly higher than IC50 for 72 h to ensure a robust assessment of the inhibitory effects. The results of the assay showed that XYA-2 had a notable inhibitory effect on colony formation in pancreatic cancer cells. Specifically, in MIA PaCa-2 cells, the inhibition of colony formation was 33%, while in PANC-1 cells, it was more pronounced at 73% (Fig. 1C). These findings indicate that XYA-2 significantly hampers the colony formation ability of pancreatic cancer cells, underscoring its potential as a therapeutic agent for pancreatic cancer treatment.

XYA-2 induces cell cycle arrest and apoptosis in pancreatic cancer cells

XYA-2 exerts an influence on the cell cycle of pancreatic cancer cells. As illustrated in Fig. 2A, treatment with 2 μM XYA-2 led to a significant increase in the number of PANC-1 cells in the G2/M phase, while concurrently reducing the number of cells in the G0/G1 phase. These results indicate that XYA-2 can effectively impede the transition of PANC-1 cells from the G2 phase to the M phase, thereby inhibiting cell proliferation.

Figure 2 XYA-2 induces cell cycle arrest and apoptosis in pancreatic cancer cells.

(A) Analysis of cell cycle distribution in PANC-1 cells treated with XYA-2 for 24 hours using propidium iodide/RNase. (B and C) Detection of apoptosis in MIA-PaCa2 and PANC-1 cells treated with compound XYA-2 at varying concentrations for 24 h using FITC-Annexin V assay. (D) Western blot tests for apoptosis and STAT3 signaling with the treatment of indicated concentrations of XYA-2.

XYA-2 also promotes apoptosis in pancreatic cancer cells. At a concentration of 5 μM, the apoptosis rate was measured as 5.48% for MIA-PaCa2 and 9.18% for PANC-1 cells. When the concentration was doubled, the apoptosis rates increased to 9.60% for MIA-PaCa2 and 12.90% for PANC-1 cells (Figs. 2B and 2C). We conducted Western blot analysis targeting apoptosis-related markers, including caspase-3, cleaved caspase-3, PARP and cleaved PARP. Our results reveal that, as the concentrations of XYA-2 increased, the protein levels of caspase-3 and PARP decreased, while the levels of cleaved PARP increased. Notably, there was no concurrent increase in cleaved caspase-3 (Fig. 2D). We considered the possibility of other caspases, such as caspase-7, exhibiting functional similarities and sharing PARP as a substrate. The observed rise in cleaved PARP provides evidence of an overall increase in apoptosis. However, the precise mechanisms governing PARP cleavage still necessitate further exploration. Furthermore, our analysis extended to STAT3 and phosphorylated STAT3, recognized targets of XYA-2 in its anti-gastric cancer activity. Our results demonstrate that XYA-2 inhibits STAT3 phosphorylation in pancreatic cancer cells (Fig. 2D). This suggests that XYA-2 may, at least in part, exert its anti-cancer effects through the modulation of apoptosis-related proteins and the inhibition of STAT3 phosphorylation in pancreatic cancer cells.

XYA-2 exhibits concentration-dependent inhibition of invasion and migration in pancreatic cancer cells

The results from the wound healing assay, depicted in Fig. 3A, underscore the concentration-dependent inhibitory effects of XYA-2 on the migratory capacity of MIA-PACA2 cells. After 24 h, the migration rates for MIA-PACA2 cells treated with 2, 5, and 10 μM XYA-2 were 22.3%, 8.6%, and 6.4%, respectively. Subsequently, at 36 h, the migration rates for MIA-PaCa2 cells exposed to 2, 5, and 10 μM XYA-2 were recorded as 41.9%, 22%, and 4.1%, respectively. Figure 3B presents the transwell assay results, highlighting XYA-2’s inhibitory effects on the invasive capability of PANC-1 and MIA-PaCa2 cells. Notably, reductions of 60% and 36% were observed in PANC-1 and MIA-PaCa2, respectively, at a concentration of 2 μM XYA-2. Importantly, these inhibitory effects on invasive ability surged significantly, reaching 99.2% for PANC-1 and 64% for MIA-PaCa2 when treated with 5 μM XYA-2. These findings affirm the dose-dependent impact of XYA-2 on the migration and invasion of pancreatic cancer cells.

Figure 3 XYA-2 exhibits concentration-dependent inhibition of invasion and migration in pancreatic cancer cells.

(A) Wound closure of MIA-PaCa2 cells treated with compound XYA-2 at indicated concentrations for 24 and 36 h. (B) Transwell assay of PANC-1 and MIA-PaCa2 cells after 48 h of treatment with compound XYA-2 at indicated concentrations.

XYA-2 as a potential inhibitor of pancreatic cancer cell progression through apoptosis induction and immune modulation

To elucidate the mechanism underlying the inhibitory effect of XYA-2 on pancreatic cancer cell progression, transcriptome analysis was conducted on cells treated with 10 μM XYA-2 and compared to control cells. In MIA-PaCa2 cells, 3,745 genes were found to be upregulated and 2,089 genes were downregulated in XYA-2-treated cells compared to the control group. Similarly, in PANC-1 cells, both upregulated and downregulated genes in XYA-2-treated cells were identified, totaling 4,324 genes. To account for any ontogenetic differences between the two cell strains, the intersection of upregulated and downregulated genes was identified separately. Ultimately, 1,933 upregulated genes and 1,158 downregulated genes were identified in XYA-2-treated cells (Fig. 4A).

Figure 4 XYA-2 as a potential inhibitor of pancreatic cancer cell progression through apoptosis induction and immune modulation.

(A) Genes that were significantly up- and down-regulated in PANC-1 and MIA-PaCa2 cells treated with XYA-2 compared to DMSO. (B) Gene set enrichment analysis of genes with abnormal expression levels in XYA-2 treatment compared to DMSO.

Following this, GSEA enrichment analysis was conducted on the dysregulated genes. Examination of hallmark gene sets revealed predominant enrichment of proliferation-related gene sets, such as “G2M checkpoint,” “E2F targets,” and “mitotic spindle,” in the DMSO treatment group. In contrast, the gene set “apoptosis” was significantly enriched in the XYA-2 treatment group (Fig. 4B). This suggests a pivotal role of XYA-2 in inducing apoptosis and impeding cell proliferation. Additionally, analysis of KEGG gene sets indicated prominent enrichment of immune-related pathways, specifically “antigen processing and presentation” and “natural killer cell-mediated cytotoxicity,” in the XYA-2 treatment group (Fig. 4B). This implies potential implications of XYA-2 in modulating immune responses against pancreatic cancer cells.

XYA-2 downregulates critical genes, associated with better survival in pancreatic cancer, and shows potential as a regulator of pancreatic cancer progression

Based on the results of the transcriptome analysis described above, we screened the top ten downregulated genes, namely ESPL1 (extra spindle pole bodies like 1, separase), PIF1 (PIF1 5′-to-3′ DNA helicase), PLK1 (polo like kinase 1), SPC24 (SPC24 component of NDC80 kinetochore complex), CCNF (cyclin F), TUBGCP5 (tubulin gamma complex component 5), KIF2C (kinesin family member 2C), KIF4A (kinesin family member 4A), TOP2A (DNA topoisomerase II alpha), and MKI67 (marker of proliferation Ki-67) (Fig. 5A). Further analysis using the public database GEPIA revealed that the expression of six genes (PLK1, SPC24, KIF2C, TOP2A, MKI67, and KIF4A) was significantly associated with overall survival in pancreatic cancer patients. Notably, low expression levels of these genes were strongly correlated with higher survival rates in pancreatic cancer patients (P-values ranging from 0.0019 to 0.036) (Fig. 5B). To validate these findings, we analyzed the expression of these genes in clinical samples of pancreatic cancer. As shown in Fig. 5B, the expression levels of these genes were significantly higher in pancreatic cancer tissue compared to adjacent normal tissue. Furthermore, existing reports suggest that these genes are involved in the development and progression of various types of cancer (Uusküla-Reimand & Wilson, 2022; Wei et al., 2021; Wu et al., 2021; Zhu et al., 2015). To further investigate the influence of XYA-2 on the expression of key genes and its role in pancreatic cancer cell development, we treated MIA-PaCa2 cells and PANC-1 cells with different concentrations of XYA-2. Subsequent qRT-PCR analysis revealed a significant downregulation of all tested genes in XYA-2-treated cells, regardless of whether the concentration used was 5 or 10 μM (Fig. 5C). These results further support the notion that XYA-2 may play a crucial role in regulating the expression of key genes involved in pancreatic cancer progression.

Figure 5 XYA-2 downregulates critical genes, associated with better survival in pancreatic cancer, and shows potential as a regulator of pancreatic cancer progression.

(A) Heatmap illustrating the top ten downregulated genes in XYA-2 treated cells. (B) The correlation between mRNA expression levels of the specified genes and overall survival time in pancreatic cancer patients (left), and the differential gene expression levels between pancreatic cancer tissues and normal pancreatic tissues (right). (C) The mRNA expression levels of the specified genes in MIA-PaCa2 and PANC-1 cells were determined by qRT-PCR after treatment with various concentrations of XYA-2.

Discussion

Pancreatic cancer is a highly metastatic disease with a poor prognosis, often diagnosed at an advanced stage when surgical intervention is not possible (Siegel et al., 2023). The median survival for untreated patients is typically only 5 to 6 months, and even with treatment, it remains less than 12 months (Rahib et al., 2014). To address this high mortality rate, researchers have developed targeted agents like erlotinib and olaparib, which target specific mutations in pancreatic cancer cells. However, resistance to these targeted drugs remains a challenge in treatment (Sherman & Beatty, 2023). In recent years, novel strategies have emerged to enhance pancreatic cancer treatment. Immunotherapy, targeting the immune system and exemplified by individualized neoantigen vaccines and checkpoint inhibitors, shows promise in boosting the immune response against pancreatic cancer cells. A recent phase I trial combining autogene cevumeran, atezolizumab, and chemotherapy successfully induced robust neoantigen-specific T cell activity, suggesting a potential avenue for improved outcomes (Rojas et al., 2023). Simultaneously, microbiota-derived metabolite indole-3-acetic acid (3-IAA) has been identified as a factor influencing chemotherapy response in pancreatic cancer. Administration of 3-IAA increased chemotherapy efficacy in mouse models, emphasizing the potential impact of microbiota interventions in cancer treatment (Tintelnot et al., 2023).

In this study, we focused on exploring the potential of a marine natural product called XYA-2 in pancreatic cancer. Our findings revealed that XYA-2 exhibited significant effects on various cellular processes in pancreatic cancer. It inhibited pancreatic cancer cell proliferation in a concentration-dependent manner. Furthermore, it influenced the transition of cancer cells during the cell cycle, leading to increased apoptosis at higher concentrations. Moreover, XYA-2 demonstrated a pronounced inhibition of migration and invasion of pancreatic cancer cells.

We further investigated the underlying molecular mechanisms by examining the modulation of critical genes and signaling pathways associated with pancreatic cancer cell proliferation. Among these, PLK1, SPC24, KIF2C, TOP2A, MKI67, and KIF4A, have been implicated in cancer development and serve as important biomarkers of proliferative activity in pancreatic cancer. SPC24, a component of the Ndc80 kinetochore complex, has been implicated in the development of various cancers. In breast cancer, SPC24 has been found to promote cancer development by regulating the PI3K/AKT pathway (Zhou et al., 2018). Studies have shown that inhibiting the expression of SPC24 effectively suppressed cell proliferation and invasion in hepatocellular carcinoma (Zhu et al., 2015). KIF2C, which is highly expressed in hepatocellular carcinoma, has been identified as a direct target of the Wnt/β-catenin pathway. It plays a critical role in mediating crosstalk between the Wnt/β-catenin and mTORC1 signaling pathways, which are involved in cancer progression and proliferation (Gan et al., 2019). Another member of the KIF2C family, KIF4A, is regulated by the transcription factor FOXM1 and has been shown to promote hepatocellular carcinoma cell proliferation (Hu et al., 2019). PLK1, a well-known player in pancreatic ductal carcinoma development, is a key regulator of cell division and has been associated with tumor growth and metastasis (Zhang et al., 2022). Targeting PLK1 has emerged as a potential therapeutic strategy for pancreatic cancer. One approach involves using superparamagnetic iron oxide nanoparticles coupled with PLK1-siRNA to specifically inhibit PLK1 expression and suppress pancreatic cancer cell proliferation (Mahajan et al., 2016). TOP2A, also known as DNA topoisomerase II alpha, plays a crucial role in DNA replication and is involved in the regulation of chromosomal structure and integrity. It has been shown to be upregulated in various cancers, including pancreatic cancer (Pei, Yin & Liu, 2018). Inhibition of TOP2A has been explored as a potential therapeutic strategy for cancer treatment, as it can disrupt DNA replication and induce cell death (Uusküla-Reimand & Wilson, 2022). MKI67, also known as Ki-67, is another proliferation-associated marker commonly used in cancer research. It is considered an essential indicator of tumor cell growth and has been associated with poor prognosis in pancreatic cancer (Uxa et al., 2021). High expression of MKI67 is indicative of increased proliferative activity and is often correlated with aggressive tumor behavior (Wu et al., 2021).

The implication of these genes and signaling pathways in cancer development highlights their potential as therapeutic targets. A more profound understanding of the exact mechanisms through which they contribute to cancer progression could offer valuable insights for devising innovative treatment strategies. Further research is imperative to explore the intricate molecular interactions, and evaluate the efficacy of XYA-2 against potential molecular targets in pancreatic cancer, both in animal models and clinical trials.

Conclusion

The discovery of XYA-2 as a potent inhibitor of pancreatic cancer cell proliferation, along with its modulation of critical pathways, represents a significant breakthrough. Delving deeper into the mechanisms of these key pathways not only enriches our understanding of XYA-2’s therapeutic potential but also positions it as a promising candidate for further translational exploration. These efforts may unveil new targets for pancreatic cancer treatment, potentially reshaping and advancing the current landscape of pancreatic cancer therapy.

Supplemental Information

Supplemental Information 1 Raw numerical data for cell line experiments (Figures 1, 2, 3 & 5).

Click here for additional data file.

Supplemental Information 2 Uncropped pictures of western blot.

Click here for additional data file.

Supplemental Information 3 MIQE Checklist.

Click here for additional data file.

We thank the use of instruments at the Shared Instrumentation Core Facility and specialists’ technical support at the Hangzhou Institute of Medicine (HIM), Chinese Academy of Sciences.

Additional Information and Declarations

Competing Interests

Author Contributions

DNA Deposition

Data Availability

The authors declare that they have no competing interests.

Xiaoqing Guan conceived and designed the experiments, analyzed the data, prepared figures and/or tables, authored or reviewed drafts of the article, and approved the final draft.

Yun Li performed the experiments, analyzed the data, prepared figures and/or tables, and approved the final draft.

Xiaodan Guan performed the experiments, analyzed the data, prepared figures and/or tables, and approved the final draft.

Linfei Fan analyzed the data, prepared figures and/or tables, and approved the final draft.

Jieer Ying conceived and designed the experiments, authored or reviewed drafts of the article, and approved the final draft.

The following information was supplied regarding the deposition of DNA sequences:

The RNA sequences of MIA-PaCa2 and PANC-1 cells with XYA-2 or DMSO treatment described here are available at GenBank BioProject: PRJNA1019926.

The following information was supplied regarding data availability:

The raw data are available in the Supplemental Files.

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
