# Peer review of "XYA-2: a marine-derived compound targeting apoptosis and multiple signaling pathways in pancreatic cancer"

_PeerJ, doi:10.7717/peerj.16805_

## Round 0.1 · original submission · Major Revisions

Please address the reviewers' comments carefully, to improve the manuscript quality.

Reviewer 1 ·

Basic reporting

This is a well-designed study. Guan et al. have examined the anticancer activity of a marine natural product XYA-2 in pancreatic cancer cell models. The results showed that XYA-2 inhibited cell viability and colony formation, induced cell cycle arrest and apoptosis, and prevented cancer cell migration and invasion. They further conducted mechanisms of action study through RNA-seq and bioinformatic analysis. It was found that XYA-2 might exert its anti-pancreatic cancer activity by modulating multiple signaling pathways and genes. In summary, this is a good paper and may be accepted for publication after making minor revisions. 

1. 2.3 Cell viability assay: the information on the plate reader used in this study should be provided. 
2. 2.5 Cell cycle arrest assay: the information on the flow cytometer should be provided. 
3. 2.6 “Apoptosis” should be “Apoptosis assay”. More importantly, the authors should double-check the treatment time in this study, 24 hours or 48 hours? Actually, in the legend for Figure 2, the authors claimed that they treated Mia-Paca2 and PANC-1 cells for 48 hours to detect cell apoptosis, which is different from the treatment time in the method section. In addition, the information on the flow cytometer should be provided.  
4. 2.7 Wound healing assay: which kind of microscope was used for recording the images in this study? 
5. 2.8 Transwell migration assay: please provide the information on the inverted microscope in this study.

Experimental design

The experiment design is reasonable and can show the required results

Validity of the findings

The theme is very interesting

Additional comments

This is a well-designed study. Guan et al. have examined the anticancer activity of a marine natural product XYA-2 in pancreatic cancer cell models. The results showed that XYA-2 inhibited cell viability and colony formation, induced cell cycle arrest and apoptosis, and prevented cancer cell migration and invasion. They further conducted mechanisms of action study through RNA-seq and bioinformatic analysis. It was found that XYA-2 might exert its anti-pancreatic cancer activity by modulating multiple signaling pathways and genes. In summary, this is a good paper and may be accepted for publication after making minor revisions. 

1. 2.3 Cell viability assay: the information on the plate reader used in this study should be provided. 
2. 2.5 Cell cycle arrest assay: the information on the flow cytometer should be provided. 
3. 2.6 “Apoptosis” should be “Apoptosis assay”. More importantly, the authors should double-check the treatment time in this study, 24 hours or 48 hours? Actually, in the legend for Figure 2, the authors claimed that they treated Mia-Paca2 and PANC-1 cells for 48 hours to detect cell apoptosis, which is different from the treatment time in the method section. In addition, the information on the flow cytometer should be provided.  
4. 2.7 Wound healing assay: which kind of microscope was used for recording the images in this study? 
5. 2.8 Transwell migration assay: please provide the information on the inverted microscope in this study.

Reviewer 2 ·

Basic reporting

no comments

Experimental design

no comments

Validity of the findings

no comments

Additional comments

Title : Marine natural product XYA-2 inhibits pancreatic cancer by targeting apoptosis and multiple signaling pathways (#91581)
The manuscript by Guan et al. is well-written and holds significant appeal for a broad readership within the pancreatic cancer research community. Their study revolves around the utilization of the marine natural product XYA-2 for targeting pancreatic cancer cells. However, there are a few comments that need to be addressed before it can be considered for publication.
Minor Comments
The manuscript's title should be revised to place a stronger emphasis on XYA-2.
Abstract
Line #23-25: Pancreatic cancer is a highly aggressive….explain about XYA-2
Line #30-32: Furthermore, RNA-seq and bioinformatics analysis were conducted to elucidate the mechanism of action of XYA-2 on pancreatic cancer cells, and the identified mechanism was confirmed through qRT-PCR.---rewrite this sentence for the better readability.
Introduction
Line # 42-45: Pancreatic cancer …authors should include latest GLOBOCAN or SEER data of 2023 for 5-year overall survival rate for pancreatic cancer.
Materials & Methods
Line # 88: concentrations of XYA-2 ranging from 0.1 to 100 …..include micromolar concentrations with all provided in manuscript.
Line # 127 : panaxadiol, were seeded into the upper chamber…..authors introduced new compound panaxadiol….hope it was a typo need to revise and rewrite the section transwell migration assay.
Line # 139 : incubation period, cells were collected…. 1 mg of total RNA. The entire sentence should be rephrased.
Line # 169- Presenting the primers utilized in the manuscript in a tabular format would be beneficial for the reader.
Line # 196- Version of the GraphPad Prism used in the manuscript should be explained.
Results
Line#232: Figure 3A…dose dependent… rewrite the sentences for readability.
Line #255- 265- Reformat the sentences.
Line #298 Immunotherapeutic therapies that target the immune system…. rewrite as Immunotherapy that target the immune system.
Line 321----337---- Incorporate the complete names of all the proteins that were initially abbreviated in the sentences.
Conclusion
The conclusion should be crafted with a greater emphasis to offer robust direction for the future utilization of XYA-2 and to delve into its translational applicability.
Major Comments
The manuscript contains numerous typographical errors that require thorough proofreading to ensure perfection.
Line #177-Line #182- Revise the methodology to include the concentrations of the primers employed in the manuscript.
Line # 344. Additionally, more studies are required…clinical trials…the entire sentence should be reframed.
Results
Proliferation data of XYA-2 are only showing 72h and 24h data, it is a requirement to show 48h data.
Rationale for the selection for 2microM concentration for colony formation assay should be explained.
It's important to thoroughly discuss or provide explanations for the selection of concentrations used in all experiments.
Treatment time not mentioned for the use of XYA-2 for RNA seq analysis and further validation using real time PCR.
To achieve translational confirmation, it's advisable to substantiate apoptosis by conducting Western blot analysis for cleaved caspase 3 and cleaved PARP.
Considering that XYA-2 was originally reported as an inhibitor of STAT proteins, it is recommended to validate the expression profile of STAT in Pancreatic cancer cells when treated with XYA-2.

Annotated reviews are not available for download in order to protect the identity of reviewers who chose to remain anonymous.

Reviewer 3 ·

Basic reporting

no comment

Experimental design

no comment

Validity of the findings

no comment

Additional comments

In this study, the authors have evaluated the anticancer efficacy of a previously identified STAT3 inhibitor, termed XYA-2 in pancreatic cancer cell lines. They have demonstrated that XYA-2 exhibited significant anticancer activity in vitro. The authors also performed RNA seq and bioinformatic analysis and found that XYA-2 could inhibit pancreatic cancer cells by inhibiting the expression of multiple genes, PLK1, SPC24, KIF2C, TOP2A, MKI67, and KIF4A, which were shown to be correlated with pancreatic cancer survival. The RNA seq results were further confirmed by qRT-PCR analysis. The following concerns need to be addressed:

1. In Line 53, the authors should be noticed that KRAS G12C mutation occurs in only 1 to 2% of pancreatic cancer while G12D and G12V mutations are more popular in this disease. In this case, pancreatic cancer patients are more likely to benefit from KRAS G12D inhibitors, such as MRTX1133.

2. In Lines 59-65, the authors have provided an informative summary of the previous studies of XYA-2 in gastric cancer cell models and mouse models, which indicated that XYA-2 exerted significant anti-gastric cancer activity in vitro and in vivo by inhibiting STAT3 phosphorylation. The question here is that the authors did not justify their further evaluation of XYA-2 in pancreatic cancer models. More specifically, why do they believe that XYA-2 may also exhibit significant efficacy in pancreatic cancer models? Is the STAT3 signaling pathway also important for pancreatic cancer development and progression?

3. In Lines 292-294, the authors claimed that both erlotinib and olaparib have been developed for targeting specific mutations in pancreatic cancer cells. This statement should be double-checked. The specific targets of erlotinib and olaparib should be introduced. It should also be clarified whether both drugs are used for clinical treatment of pancreatic cancer patients.

---

## Round 0.2 · accepted · Accept

After carefully checking, this revised version is good enough and could be accepted.

Reviewer 2 ·

Basic reporting

no comments

Experimental design

no comments

Validity of the findings

no comments

Additional comments

Thanks to authors for completing the required reviews.
Best wishes for future endeavors.

Reviewer 3 ·

Basic reporting

no comment

Experimental design

no comment

Validity of the findings

no comment

Additional comments

The authors have addressed all my concerns, and the manuscript has been greatly improved. The article is recommended for acceptance.